# Facile Fabrication of MnCo_2_O_4_/NiO Flower-Like Nanostructure Composites with Improved Energy Storage Capacity for High-Performance Supercapacitors

**DOI:** 10.3390/nano11061424

**Published:** 2021-05-28

**Authors:** Sangaraju Sambasivam, K. V. G. Raghavendra, Anil Kumar Yedluri, Hammad Mueen Arbi, Venkatesha Narayanaswamy, Chandu V. V. Muralee Gopi, Byung-Chun Choi, Hee-Je Kim, Salem Alzahmi, Ihab M. Obaidat

**Affiliations:** 1Department of Physics, United Arab Emirates University, Al Ain 15551, United Arab Emirates; sambaphy@gmail.com (S.S.); hammad.arbi@yahoo.com (H.M.A.); venkateshnrn@gmail.com (V.N.); 2School of Electrical Engineering, Pusan National University, Busan 46241, Korea; kvgraghavendra999@gmail.com (K.V.G.R.); yedluri.anil@gmail.com (A.K.Y.); heeje@pusan.ac.kr (H.-J.K.); 3Department of Electrical Engineering, University of Sharjah, Sharjah P.O. Box 27272, United Arab Emirates; naga5673@gmail.com; 4Department of Physics, Pukyong National University, Busan 608737, Korea; bcchoi@pknu.ac.kr; 5Department of Chemical & Petroleum Engineering, United Arab Emirates University, Al Ain 15551, United Arab Emirates; s.alzahmi@uaeu.ac.ae

**Keywords:** MnCo_2_O_4_/NiO, electrode, supercapacitor, nanostructure, chemical bath deposition, specific capacity

## Abstract

Over the past few decades, the application of new novel materials in energy storage system has seen excellent development. We report a novel MnCo_2_O_4_/NiO nanostructure prepared by a simplistic chemical bath deposition method and employed it as a binder free electrode in the supercapacitor. The synergistic attraction from a high density of active sites, better transportation of ion diffusion and super-most electrical transportation, which deliver boost electrochemical activities. X-ray diffraction, field-emission scanning electron microscopy, and X-ray photoelectron spectroscopy have been used to investigate the crystallinity, morphology, and elemental composition of the as-synthesized precursors, respectively. Cyclic voltammetry, galvanostatic charge/discharge, and electron impedance spectroscopy have been employed to investigate the electrochemical properties. The unique nanoparticle structures delivered additional well-organized pathways for the swift mobility of electrons and ions. The as-prepared binder-free MnCo_2_O_4_/NiO nanocomposite electrode has a high specific capacity of 453.3 C g^−1^ at 1 Ag^−1^, and an excellent cycling reliability of 91.89 percent even after 4000 cycles, which are significantly higher than bare MnCo_2_O_4_ and NiO electrodes. Finally, these results disclose that the as-fabricated MnCo_2_O_4_/NiO electrode could be a favored-like electrode material holds substantial potential and supreme option for efficient supercapacitor and their energy storage-related applications.

## 1. Introduction

With the advancements in the world’s energy sector, there has been a rapid enhancement in the evolution of various energy storage devices, namely rechargeable batteries and supercapacitors (SCs). Some of the most attractive features of SCs that make them unique are the increased specific power and the predominant charge/discharge cycles with excellent life span [1,2,3,4]. SCs are categorized into electrochemical double-layer capacitors (EDLCs) and pseudocapacitors. EDLCs implement a non-faradaic process where the charge separation takes place at the electrode/electrolyte interface and leads to the charge storage. Contrarily, pseudocapacitors store charge by the faradaic process at the surface of the electrode. In general, the electrode material is critical in defining the performance of SCs [5,6,7,8,9]. Hence, it is highly essential to fabricate high-performance SCs by developing advanced electrode materials with a unique morphology [10,11,12].

Recently, mixed metal oxides based on first-row transition metals such as NiO, Co_3_O_4_, NiCo_2_O_4_, MnCo_2_O_4_, and MnO_2_ (TMOs) have attracted much attention for both battery-type and pseudocapacitive SC applications due to their enhanced capacities [5,6,7,8,9]. Due to their strong electrochemical conductivity, rapid redox activity, high theoretical power, and low cost, pseudocapacitive electrodes (e.g., MnO_2_ and RuO_2_), and battery-type electrode materials exhibit higher energy storage [13,14,15,16]. MnCo_2_O_4_ has the best physicochemical and electrochemical properties of all the TMOs because cobalt has a higher oxidation potential and manganese can hold a huge number of electrons, resulting in a higher power [17,18]. However, the MnCo_2_O_4_ exhibits lower electrical conductivity, which is an obstacle for its practical application. Various MnCo_2_O_4_ composites, such as C@MnCo_2_O_4_, MnCo_2_O_4_@MnO_2_, MnCo_2_O_4_@Ni(OH)_2_, nanoflakes, and MCO/graphene nanoplates, have been widely prepared and used for SC applications to improve the energy storage efficiency of MnCo_2_O_4_ material [17,18,19]. Nickel oxide (NiO) on the other hand, due to its strong theoretical basic capacitance, excellent redox ability, and good thermal stability, is a promising electrode material [20]. However, the practical application of NiO is limited due to its poor cyclic stability [21]. By utilizing the advantages of both the MnCo_2_O_4_ and NiO materials, the MnCo_2_O_4_ is integrated with NiO in the present study.

In this work, the chemical bath deposition (CBD) method was employed to fabricate a novel flower-like MnCo_2_O_4_/NiO on Ni foam surface to be effectively used as a potential electrode for supercapacitors. CBD method is a cost-effective and straightforward deposition method where the deposition is possible for a wider area. The as-synthesized precursors had a high-energy storage capacity of 453.3 C g^−1^ at 1 A g^−1^ and strong cycling stability. Due to the faster rate of electrolytic diffusion and faster electron paths, the composite electrode has a greater energy storage capacity. As a result, the MnCo_2_O_4_/NiO nanocomposite electrode may be a potential supercapacitor electrode material.

## 2. Experimental Details

### 2.1. Materials

The nanocomposites of MnCo_2_O_4_/NiO fabricated using high pure analytical grade reagents and all the chemicals were purchased from Sigma-Aldrich. Cobalt nitrate hexahydrate (Co(NO_3_)_2_·6H_2_O), manganese acetate (Mn(CH_3_CO_2_)_2_·4H_2_O), ammonium fluoride (NH_4_F), aqueous ammonia (NH_4_OH), nickel nitrate (Ni(NO_3_)_2_·6H_2_O), CH_4_N_2_O, thioacetamide (C_2_H_5_NS), hydrochloric acid (HCl), and potassium hydroxide (KOH).

### 2.2. Fabrication of the Flower-Like MnCo_2_O_4_/NiO Electrode

The fabrication of electrode supercapacitor started by cleaning nickel foams (1 × 2 cm^2^) in 2 M HCl followed by acetone, ethanol, and distilled (DI) water sonication for 15 min. The MnCo_2_O_4_/NiO composite was fabricated by implementing the CBD technique. Typically, 0.2 M Co(NO_3_)_2_·6H_2_O, 0.1 M Mn(CH_3_CO_2_)·4H_2_O, 0.2 M Ni(NO_3_)_2_·6H_2_O, 0.4 M CH_4_N_2_O, 0.24 M NH_3_, and 0.4 M NH_4_F were mixed in 80 mL of DI water and kept under magnetic stirring for about 30 min. The washed nickel foams were soaked in the chemical solution prepared above and dried at 70 °C for 12 h. The foams were washed again with ethanol and DI water before being dried in the oven at 70 °C for another 10 h. A similar technique was used to make MnCo_2_O_4_ and NiO, but without the inclusion of Ni sources for MnCo_2_O_4_ and Mn and Co sources for NiO. MCO and MCO/NiO are the abbreviations for MnCo_2_O_4_ and MnCo_2_O_4_/NiO, respectively. On Ni foam substrates, the average weights of MCO, NiO, and MCO/NiO electroactive materials were found to be 4.18, 4.37, and 4.55 mg cm^−2^, respectively.

### 2.3. Characterization

Powder X-ray diffraction (XRD, D/Max-2400 Rigaku, Pusan National University, Busan, Korea), energy-dispersive X-ray spectroscopy (XPS, VG Scientific ESCALAB 250, Busan center of KBSI, Busan, Korea), and scanning electron microscopy (FE-SEM, S-2400, Hitachi, Pusan National University, Busan, Korea) were used to investigate the phase structure, the elemental composition and the oxidation states, and the morphology of the composite electrode, respectively.

### 2.4. Electrochemical Measurements Using Cyclic Voltammetry

The electrochemical experiments were carried out on a BioLogic–SP150 workstation in an aqueous 3 M KOH electrolyte solution. Pt wire was used as the counter electrode, Ag/AgCl was used as the reference electrode, and the synthesized precursors were used as the operating electrodes in a three-electrode setup.

In the voltage range of 0–0.55 V, cyclic voltammetry (CV) measurements were taken at different scan frequencies (10, 20, 50, and 75 mV s^−1^). A galvanostatic charge–discharge (GCD) test was also carried out to determine the specific capacity (*C_S_*_,_ C g^−1^) using Equation (1) [22]:(1)$$\;C_{S}=\frac{i\times \Delta t}{m}\;$$

Here, *C_s_* denotes the specific capacity of C g^−1^; *i* denote the discharge current in ampere (A); *t* denotes the difference in discharge time in seconds; *m* denotes the mass (g) of active materials.

## 3. Results and Discussion

XRD was used to analyze the composition and phase structure of the as-synthesized composite. As can be seen in Figure 1, the XRD patterns of the as-prepared MCO precursor illustrated the corresponding peaks at (311), (400), and (511) that were correlated with the JCPDS: 23-1237 [23]. At the same time, the NiO precursor peaks were observed at (111), (200), and (311) and well matched with the JCPDS: 78-0429 [24]. Further, the MCO/NiO composite strongly displayed all the peaks of both MCO and NiO samples, which suggest that the as-synthesized composite has pure phases and is crystalline in nature.

The XPS analysis verified the chemical structure and oxidation states of the as-fabricated electrode, and the findings are seen in Figure 2. The elements of Ni 2p, Co 2p, and Mn 2p were identified in the MCO/NiO electrode based on the XPS survey spectrum (Figure 2a). C and O are two components that are commonly found in the air. The two dominant peaks of Mn 2p_3/2_ at a binding energy of 643.7 eV and Mn 2p_1/2_ at a binding energy of 652.2 eV in the Mn 2p XPS range are ascribed to the Mn^2+^ binding energy, as seen in Figure 2b. The peaks at 777.84 eV and 787.15 eV in the Co 2p high-resolution range seen in Figure 2c belong to Co 2p_3/2_ and Co 2p_1/2_, respectively [25]. Further, Ni 2p peaks are illustrated in Figure 2d for Ni 2p_1/2_ and Ni 2p_3/2_ at 875.2 eV and 856.4 eV, respectively.

Furthermore, the morphology of the as-prepared electrodes was investigated by SEM characterization. Figure 3a,a1,a2 depicts the SEM images of MCO at different magnifications, and nano rice-like morphology is observed in the images due to the interlinking of the nanoparticles. Figure 3b,b1,b2 shows SEM photographs of NiO that display the integration of several uniformly sized nanosheets with absolute coherence. SEM images of MCO/NiO nanoparticles at various magnifications, where flower-like nanoparticles are densely deposited on the Ni foam substrate, are seen in Figure 3c,c1,c2. The full interaction between electrode materials and electrolytes and the transportation of electrons during charge and discharge cycles are aided by this special arrangement, which improves the electrochemical efficiency of the MCO/NiO electrode as-prepared. This unusual configuration benefits complete interaction between electrode materials and electrolytes and electron transportation during charge and discharge cycles, enhancing the electrochemical stability of the as-prepared MCO/NiO electrode.

CV, GCD, and electrochemical impedance spectroscopy (EIS, 0.01–100 kHz) experiments were performed with a three-electrode system in an aqueous 3 M KOH electrolyte to investigate the electrochemical activity of the as-prepared electrodes. The CV curves of the as-prepared electrodes at a scan rate of 10 mV s^−1^ with a potential window from 0.0 to 0.55 V are shown in Figure 4a. As seen in these CV plots, each one of the as-prepared three electrodes (MCO, NiO, and MCO/NiO) delivered a pair of redox peaks with a higher current response, signaling battery-type activity. Furthermore, when compared to bare MCO and NiO electrodes, the flower-like MCO/NiO electrode exhibited the highest peak current and loop field, suggesting a significant improvement in electrochemical Faradaic reaction kinetics and basic power. The MCO, NiO, and MCO/NiO electrodes in Figure 4b–d were obtained at the scan rates of 10, 20, 50, and 75 mV s^−1^, respectively. This study revealed that there were two pairs of redox peaks with an enhanced current response, indicating that the battery-type electrode exhibits reversible faradaic activity. Furthermore, as the scan speeds increased, the anodic and cathodic peaks changed to more positive and negative potential areas, owing to the excellent ion diffusion rate and lower internal resistance of the electrode content during redox reactions.

In addition, GCD tests were performed at various current densities with a potential ranging from 0 to 0.48 V to compare the MCO/NiO electrode’s high capacity output with those of the other electrodes. The GCD plots of the MCO, NiO, and MCO/NiO electrodes at 5 A g^−1^ are shown in Figure 5a. The non-capacitive Faradaic redox battery-type properties of both electrodes are visible on the GCD plateaus, which are distinct from the inverted “V” shapes of EDLC materials and hence comply with the CV performance. Owing to the synergistic effect of two-material electrodes, the MCO/NiO electrode had a longer charge–discharge time cycle than that of each of MCO and NiO electrodes. The GCD plots of the MCO, NiO, and MCO/NiO at different current densities of 1, 2, 5, 7, and 10 A g^−1^ are shown in Figure 5b–d. 

The GCD plateaus show that all samples have a battery-like redox activity with non-linear symmetric charge–discharge intervals, demonstrating the material’s excellent reversibility. Figure 5e shows the real capacity values as a function of the current density based on Equation (1). At different current densities of 1, 2, 5, 7, and 10 A g^−1^, the MCO/NiO electroactive material exhibited specific capacity values of 453.3, 449.6, 448.1, 444.2, and 438.8 C g^−1^, respectively, which were significantly higher than the corresponding specific capacity values of bare MCO and NiO. EIS calibrations were also used to estimate the internal resistance and conductivity of the prepared MCO, NiO, and MCO/NiO electrodes. Figure 5f shows a Nyquist plot with the same sequence resistance as the intercept with the *x*-axis (R_S_). The charge transfer resistance is shown by the semicircular curve in the high-frequency field (Rct). The fast transportation of ions/electrons is depicted by the straight line in the low-frequency field [26,27]. The equivalent circuit, shown in the inset of Figure 5f, was used to fit the Nyquist plots. The MCO/NiO electrode exhibited lower R_ct_ = 0.08 Ω cm^2^ and R_S_ = 0.19 Ω cm^2^ values than those of the MCO (R_ct_ = 0.3 Ω cm^2^ and R_S_ = 0.28 Ω cm^2^) and of NiO (R_ct_ = 0.5 Ω cm^2^ and R_S_ = 0.34 Ω cm^2^) electrodes suggesting the enhanced electrical conductivity and efficient electron transfer at the interface of the electrode/electrolyte in the MCO/NiO electrode. Furthermore, when compared to bare MCO and NiO electrodes in the low-frequency field, the flower-like MCO/NiO electrode exhibited a slope near the vertical axis, indicating a superior energy storage capacitive activity in the composite electrode. As a result, the MCO/NiO electrode with low diffusion resistance, R_S_, and Rct values is verified as a significant contributor to superior supercapacitor efficiency. More specifically, in practical supercapacitor systems, the cycling reliability of an electroactive material is critical. As shown in Figure 5g, the magnified view of the Nyquist plots for three electrode materials. It could be clear shows that the high frequency parts of the EIS profiles and obviously see them impedances of three electrode materials. Figure 5h shows the cycling performance of the MCO/NiO electrode over a period of 4000 cycles at a current density of 5 A g^−1^ (Figure 5h). The flower-like MCO/NiO electrode demonstrated excellent cycling stability of 91.89 percent over 4000 cycles. The activation mechanism of both materials through deep penetration of electrolyte ions into their interior components is responsible for the composite electrode’s outstanding cycling stability [28,29,30,31].

As shown in Figure 6a,b, even after a long term cycling stability analysis, as depicted in the SEM image and XRD pattern, flower-like nanostructure still exist in the MCO/NiO composite, revealing that there was no noticeable changes in the morphology and phase of the material. Such excellent cycling stability of the MCO/NiO material could be attributed to the gradual penetration of electrolyte ions into the electroactive material, which provides to the higher activation of the materials.

## 4. Conclusions

Precisely, flower-like nanoparticles of MCO/NiO electroactive material were synthesized on nickel foam via a cost-effective chemical bath deposition procedure. For effective supercapacitor use, the as-synthesized precursor was used as a binder-free battery style electrode material. Furthermore, the as-prepared MCO/NiO electrode demonstrated outstanding electrochemical activity, with a specific potential of 453.3 C g^−1^ at 1 Ag^−1^, and cycling reliability of 91.89 percent over 4000 cycles, all of which were significantly higher than the bare MCO and NiO electrodes. These improved electrochemical properties were attributed to the MCO/NiO nanocomposite’s superior quantity of active sites, which resulted in a larger contact surface area with the electrolyte and faster redox reactions. The outcomes hint at improving the battery-type cathode with the outstanding electrochemical properties for future high-performance supercapacitors.

## Figures and Tables

**Figure 1 nanomaterials-11-01424-f001:**
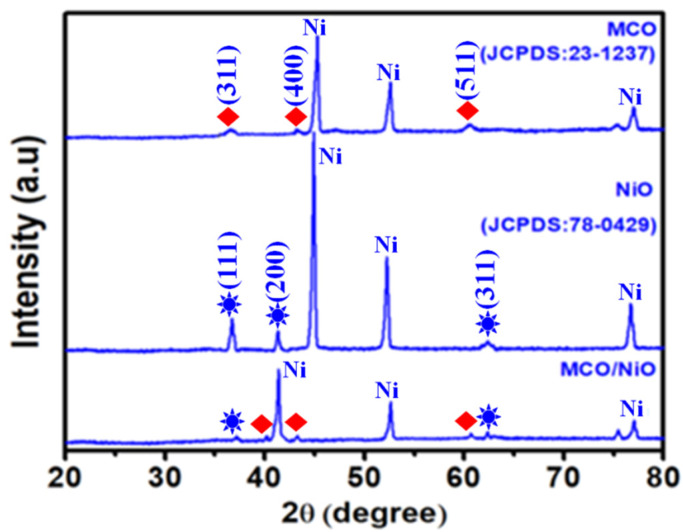
Powder X-ray diffraction pattern of the MCO, NiO, and MCO/NiO electrode materials deposited on Ni foam.

**Figure 2 nanomaterials-11-01424-f002:**
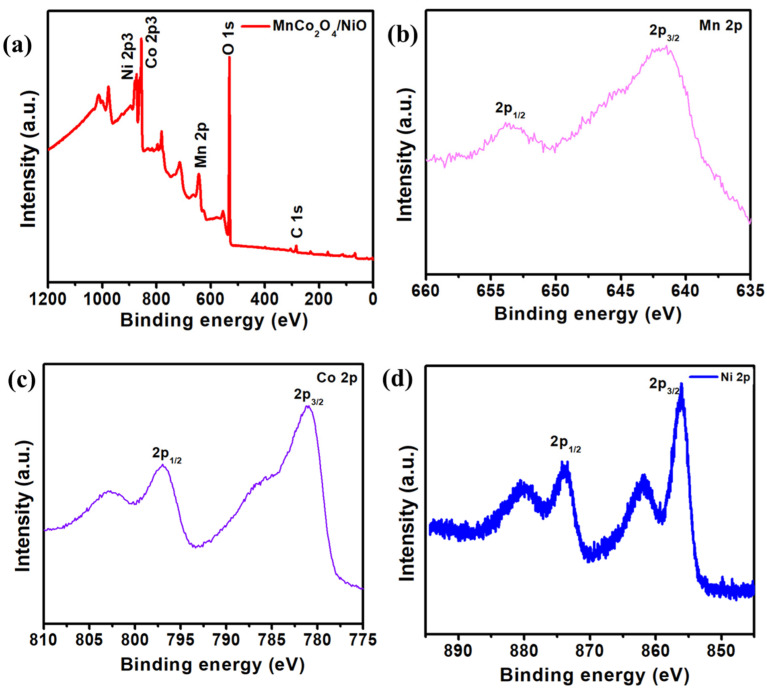
(**a**) XPS survey spectrum of the MCO/NiO composite. High resolution XPS spectra of the (**b**) Mn 2p, (**c**) Co 2p, and (**d**) Ni 2p.

**Figure 3 nanomaterials-11-01424-f003:**
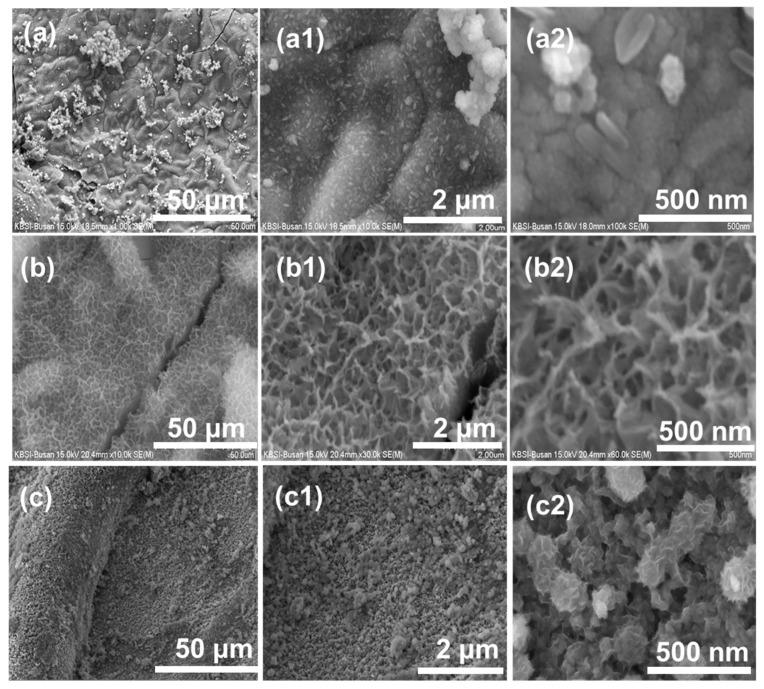
FE-SEM images (**a**,**a1**,**a2**) of MCO, (**b**,**b1**,**b2**) of NiO, and (**c**,**c1**,**c2**) of MCO/NiO under different magnifications.

**Figure 4 nanomaterials-11-01424-f004:**
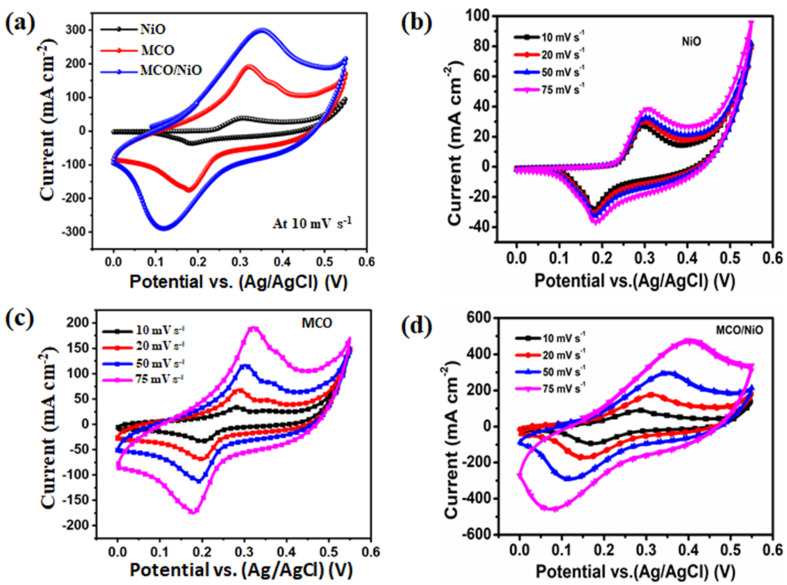
(**a**) Comparison of CV curves of MCO, NiO, and MCO/NiO electrodes at a scan rate of 10 mV s^−1^, CV curves of the (**b**) NiO, (**c**) MCO, and (**d**) MCO/NiO electrodes at various scan rates.

**Figure 5 nanomaterials-11-01424-f005:**
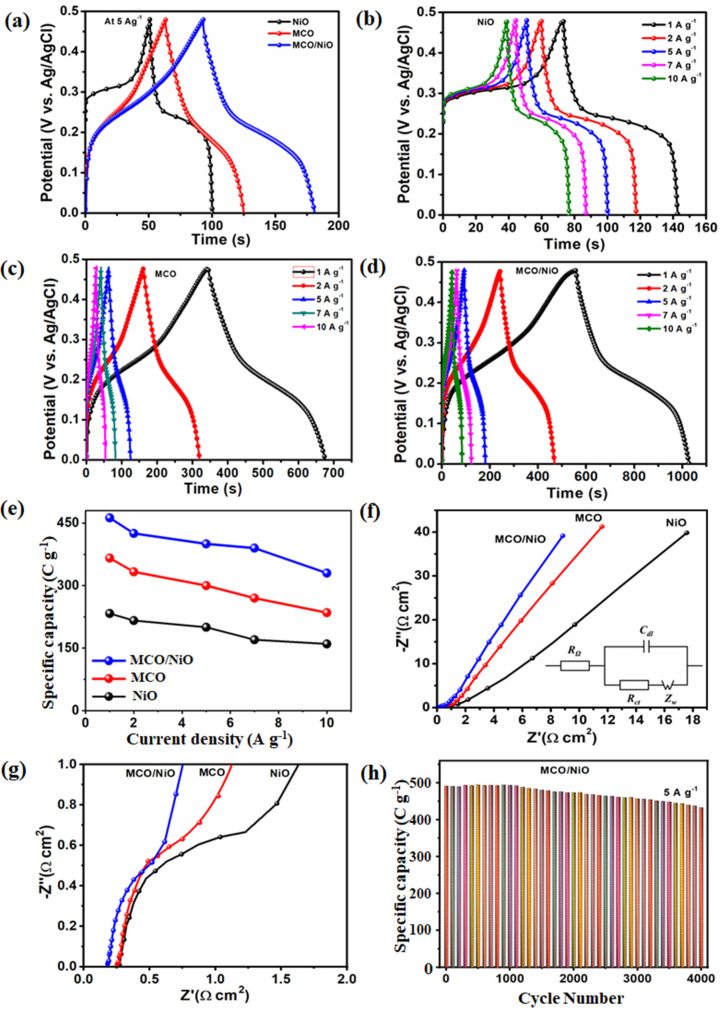
(**a**) Comparison of GCD plots for MCO, NiO, and MCO/NiO at 1 Ag^−1^, GCD plots of (**b**) NiO, (**c**) MCO, and (**d**) MCO/NiO at various current densities and (**e**) specific capacity values of MCO, NiO, and MCO/NiO electrodes with respect to current density. (**f**) EIS of MCO, NiO and MCO/NiO electrodes. (**g**) The magnified view of the Nyquist curve of MCO/NiO electrode. (**h**) Cycling stability of the MCO/NiO electrode. The magnified view of the Nyquist curves.

**Figure 6 nanomaterials-11-01424-f006:**
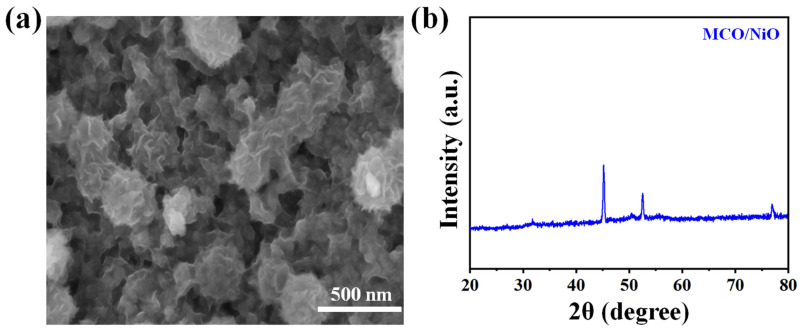
After cycling stability test, (**a**) SEM, and (**b**) XRD of the MCO/NiO composite electrode material.

## Data Availability

Not applicable.

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
