# Peer review of "Facile Fabrication of MnCo2O4/NiO Flower-Like Nanostructure Composites with Improved Energy Storage Capacity for High-Performance Supercapacitors"

_nanomaterials, 2021, doi:10.3390/nano11061424_

Round 1

Reviewer 1 Report

1) First of all, “novel” word should be carefully used in the title of the manuscript, because a hierarchical flower-like metal oxides are not a new system any more as the electrode materials for high-performance supercapacitor applications.

2) Except for the general descriptions regarding the electrode materials, such as mixed transition-metal hydroxides, more recent a lot of the references concerning the hierarchical flower-like metal oxide-based electrode materials should be introduced and compared each other.

3) In Fig. 3, importantly what is the hierarchical flower-like morphologies, which are not clear? It should be clarified in detail.

4) Except for a three-electrode setup, for a practical application, the 2-electrode based electrochemical analysis should be carried out and discussed.

5) Importantly, the energy and power densities should be studied, and the related references should be also cited in this manuscript.

Author Response

attached the review comments

Reviewer 2 Report

Freestanding supercapacitor electrode based MnCo2O4/NiO on Ni foam was prepared via a chemical bath deposition method. The as-prepared electrode demonstrated enhanced electrochemical charge storage capability than the pure MnCo2O4 and NiO on Ni foams. The material also exhibited excellent cycling stability upon 4000 cycles. This manuscript could be published on Nanomaterials after addressing the following questions.

  1. The anodic current curves of MCO in Figure 4a and 50 mV/s and 75 mV/s in Figure 4c did not reach to 5.5 V vs. Ag/AgCl. It looks they were cut off at ca. 5.2 V vs. Ag/AgCl. Authors might need to explain on this.
  2. For supercapacitors, the unit of specific capacitances should be termed in C/g rather than mAh/g. mAh/g is typically used for batteries.
  3. The high frequency parts of the EIS profiles should be magnified to clearly see the impedances of three electrode materials.
  4. It’s not appropriate to calculate the specific capacitance of pseudocapacitive materials via the discharging curves, especially for those with non-linear charging and discharging curves, as shown in this work. The area of cathodic curves in the CV should be used to calculate the capacitances.
  5. It would be good to use either different shapes or different colors of the heart shape to label the two phases of MCO and NiO in the XRD patterns (Figure 1).
  6. SEM and XRD of the electrode material after cycling stability test should be provided to see how the morphology and materials phase change with the continuous charging and discharging process.

Author Response

attached the response to review

Round 2

Reviewer 1 Report

None.

Reviewer 2 Report

The authors have addressed the issues in the revised version. The manuscript can be accepted as it is.